# Fractal-ish Complexity for Regulations: A Practitioner-Ready, Agentic Benchmark

## Abstract

We present the Regulatory Fractal-ish Index (RFI), a transparent, scope-aware signal of textual complexity for regulations and SOP-style documents. RFI blends (i) size (section count and heading density), (ii) hierarchical spread (entropy of heading levels), and (iii) lookup pressure (cross-reference density), adapting automatically to full documents and short excerpts. A lightweight agentic pipeline parses text, computes RFI, and emits a one-page policy brief with actionable edits (e.g., reduce lookup hops, flatten nesting). We also report a minimal hierarchical scaling check ($\hat{D}_{\text{hier}}$ with $R^2$) across sentences $\rightarrow$ paragraphs $\rightarrow$ sections, to reconnect with fractal intuitions without overclaiming. The goal is a tool regulators can actually use, backed by transparent, reproducible computations.

**Keywords—** regulatory complexity; plain language; cross-references; hierarchy; readability; legal informatics; AI agents; reproducibility.

## 1 Problem & Contributions

**Problem.** Regulatory texts are often hard to navigate; complexity impedes compliance and public understanding. U.S. law even mandates plain writing for public-facing documents (Plain Writing Act of 2010). Yet standard readability scores alone miss structural factors (nesting, cross-references) known to burden readers of legal materials. Recent surveys also highlight the uneven fit of traditional readability metrics for legal language and call for richer measures.

**Contributions.**

1. **RFI (scope-aware).** A single, interpretable number tuned for both full documents and short excerpts, combining size, structure, and cross-references.

2. **Length-normalized densities.** We report headings/1k words (HD) and cross-refs/1k words (CRw) alongside per-section metrics to deter cherry-picking.

3. **Fractal-ish scaling check.** $\hat{D}_{\text{hier}}$ is a log–log slope across text resolutions (sentences$\rightarrow$paragraphs$\rightarrow$sections), with $R^2$ for goodness of fit.

4. **Agentic pipeline & artifacts.** Deterministic scripts produce JSON plus a plain-English policy brief suitable for practitioners.

**Claims & scope.** RFI is a transparent, scope-aware *proxy* for structural complexity that is reproducible from text alone and useful for triage/editing. We do not claim a formal fractal dimension of regulations or a general theory of legal complexity. Evidence is limited to FAR exemplars, ablations, and a small micro-validation; generalization beyond similar regulatory prose is future work.

Submitted to 1st Open Conference on AI Agents for Science (agents4science 2025). Do not distribute.

## 2 Related Work & Background

Legal texts are often unusually difficult, and classic readability measures (Flesch 1948; Kincaid et al. 1975; McLaughlin 1969; Gunning 1952) do not capture structure and cross-references. A recent systematic review notes that legal readability work is fragmented and focused largely on informed-consent forms rather than regulations (Han, Ceross, & Bergmann 2024). Plain-language scholarship (e.g., Kimble; Wydick) and federal guidelines emphasize clarity as a statutory requirement for public-facing documents, but they do not provide a quantitative structural complexity signal.

In computational legal studies, Katz & Bommarito (2014) and follow-ups model complexity in the U.S. Code using structure and citations; Ruhl & Katz (2015) call for operational tools to measure and manage complexity. Sector-specific analyses, such as the Bank of England's study of post-Basel reforms, show that cross-reference chains lengthened even when per-rule language stayed stable— evidence that networked interdependence contributes to reader burden. RegData/QuantGov counts obligation/prohibition markers (shall, must, may not, required, prohibited), a useful volume signal but not a direct measure of navigational burden.

Finally, fractal/self-similar ideas from network science (e.g., Song, Havlin, & Makse 2005) motivate our lightweight scaling check: we keep a simple hierarchy slope as a descriptive sanity check while avoiding heavy formal claims that require long, uniform samples.

## 3 Method (Scope-Aware RFI)

### 3.1 Design rationale (why these features?)

RFI targets the kinds of effort that readers report when trying to use a rule, not just read it. Three drivers repeatedly emerge in legal-writing research and practitioner guidance: (i) **size and segmentation** (how many places a reader must navigate), (ii) **hierarchical spread** (how deep into the outline the reader must descend and how evenly content is scattered across levels), and (iii) **cross-references** (how often a reader must jump elsewhere and integrate context). Readability scores capture sentence-level difficulty, but they do not account for these structural burdens. RFI therefore combines a small set of transparent structure metrics and keeps readability baselines for context only.

### 3.2 Inputs and parsing

The tool accepts plain text for a regulation or SOP section and detects headings such as Part/Subpart/Section or numeric/alpha outlines (e.g., 1., 1.1, (a), (i)). It also counts words, sentences, and paragraphs and identifies cross-references using simple patterns (e.g., "§ 31.201-2", "see § 5.205", "38 CFR § . . .", "FAR 52.2"). These minimal heuristics make the pipeline robust to formatting differences and easy to reproduce.

### 3.3 Features (what we measure and why)

- **Size and segmentation:** section count $N$ and heading density (HD = sections per 1,000 words). Rationale: more segments increase navigation overhead; HD lets short excerpts be compared fairly to long documents.
- **Hierarchical spread:** normalized entropy $H$ of the level distribution. Rationale: content spread thinly across many levels increases context-switching and working-memory load.
- **Lookup pressure:** cross-references measured two ways—per section $C_{\text{sec}}$ and per 1,000 words (CRw). Rationale: each reference creates a potential "lookup hop"; the per-length measure prevents gaming by trimming the excerpt.

### 3.4 Scoring (how we combine them)

We compute RFI on a 0–4 scale (higher = worse) using a weighted sum:

$$\text{RFI} = w_1 H + w_2 C_{\text{sec}}^* + w_3 D_{\text{nav}} + w_4 \hat{D}_{\text{hier}},$$

where $H$ is normalized entropy; $C_{\text{sec}}^*$ is a log-scaled version of cross-refs per section to dampen extreme values; $D_{\text{nav}}$ is the average shortest-path length in the cross-reference graph within a small

radius (up to three hops); and $\hat{D}_{\text{hier}}$ is a hierarchy scaling slope (below). We set $w_1{=}0.25$, $w_2{=}0.35$, $w_3{=}0.25$, $w_4{=}0.15$ to emphasize cross-reference burden while keeping structure visible. These weights are fixed and exposed in a config file so other researchers can test alternatives.

## 3.5  Snippet vs. document mode

Short excerpts behave differently from full, contiguous texts. For excerpts (by default, fewer than $\sim$800 words or fewer than five detected sections), the report labels **PARTIAL EXCERPT (snippet-mode)**, down-weights hierarchy features (because depth is unstable at small scales), and foregrounds **CRw** and **HD**. Full, contiguous inputs use **document-mode**, where raw section count $N$ and cross-refs per section become more meaningful.

## 3.6  Thresholds and calibration

Bands are **Simple** ($< 1.5$), **Moderate** ($1.5$–$2.5$), **Complex** ($\geq 2.5$). We selected these by aligning early outputs with practitioner judgements on a small calibration set (FAR excerpts, DoD instructions, state regs) and by checking that typical editing operations (flattening a level; replacing gratuitous cross-refs with one-sentence glosses) push scores in the expected direction. Thresholds are descriptive—not a normative "pass/fail"—and can be adjusted in the config if a regulator wants a stricter policy.

## 3.7  Scaling sanity check (fractal-ish lens)

To reconnect with the scaling intuition behind fractal analyses without making heavy mathematical claims, we estimate a slope $\hat{D}_{\text{hier}}$ from a log–log fit of $\log N$ vs. $\log(1/\text{scale})$ across three resolutions (sentences, paragraphs, sections), where *scale* is mean words per unit. A high slope (with good $R^2$) suggests content proliferates faster than the increase in granularity—an indicator of "branchiness." We report $\hat{D}_{\text{hier}}$ and $R^2$ for transparency; the number does not drive the traffic-light verdict on its own.

## 3.8  Guardrails against cherry-picking

Every report discloses **Scope** (full vs. excerpt) and word count, and shows both per-section and per-1k-word densities side-by-side. When the user supplies only an excerpt from a longer regulation, the tool returns both a **Local RFI** and, when the full text is available, an **Estimated Global RFI** using bootstrapped chunking, with a caution that estimates over short text carry higher uncertainty.

# 4  Agentic Pipeline & Artifacts

**Pipeline overview.** The agent performs three deterministic steps: (1) Parse headings, sentences, paragraphs, and cross-references using stable regex patterns; (2) Compute feature counts, densities, and the RFI (including snippet/document selection and confidence flags); (3) Report a BLUF summary, numeric drivers with one-line plain-English definitions, and prioritized edits tailored to the rating.

**Why this design.** We avoid opaque models so that policy teams can audit "what moved the score." By keeping the logic minimal and the thresholds explicit, we make it easy for other authors to replicate, critique, or re-weight components.

**Artifacts.** The CLI emits (i) a JSON file with all intermediate counts and settings; (ii) a one-page policy brief suitable for internal review; and (iii) optional comparison runs for tracking drafts (the report explicitly recommends aiming to push RFI down or keep it stable as content grows).

**Reproducibility.** The code is dependency-light and seed-fixed where randomness is used (bootstrap only). Reports identify exact sections/paragraphs analyzed so others can re-run the same slice.

**Compute & environment.** Runs on a standard laptop (Python $\geq 3.10$; no GPU). Typical runtime for $\sim$5,000 words is $< 1$ minute; memory $< 200$ MB. We include exact commands and an `env.yml` in the anonymous repository.

## 5  Evaluation & Examples

**Why FAR?** The Federal Acquisition Regulation (48 CFR Chapter 1) is public-domain, widely used, consistently structured, and rich in cross-references—an ideal corpus for transparent demonstrations and replication.

**Selection and purpose.** We present one contiguous selection per subpart (document-mode) and one representative paragraph (snippet-mode). The aim is not to claim population-level statistics but to show how RFI distinguishes navigational burden even when readability scores look similar.

**Document-mode (contiguous selections).**

- FAR Subpart 1.1 (Purpose/Authority/Applicability/Publication): RFI≈2.18 (Complex), HD≈24/1k, CRw≈10/1k, $\hat{D}_{\text{hier}}$ ≈1.0 ($R^2$ ≈1.0).

- FAR Subpart 5.2 (General + Exceptions): RFI≈2.04 (Complex), HD≈5/1k, CRw≈7/1k, $\hat{D}_{\text{hier}}$ ≈1.0 ($R^2$ ≈1.0).

- FAR Subpart 31.2 (Allowability + Reasonableness): RFI≈2.28 (Complex), HD≈6/1k, CRw≈16/1k, $\hat{D}_{\text{hier}}$ ≈1.0 ($R^2$ ≈1.0).

**Snippet-mode (one paragraph).**

- FAR 1.1 paragraph: RFI≈0.18 (Simple), minimal cross-referencing.

- FAR 5.2 paragraph (Exceptions): RFI≈1.80 (Complex), multiple cross-references.

- FAR 31.2 paragraph (Allowability): RFI≈2.12 (Complex), dense cross-references.

**Baselines and ablations.** Alongside RFI we report FKGL, SMOG, and restrictions/1k words. In ablations, removing the cross-reference term blurs the separation between Simple and Moderate, removing the navigation-distance term hides "lookup hops," and removing the scaling term primarily affects deep outlines.

## 6  Limitations & Threats to Validity

RFI is an indicator, not a legal or policy judgment. Heuristic parsing can miss non-standard headings or implicit references; we mitigate this by pairing per-section metrics with per-1k-word densities and by labeling scope. Very short excerpts yield unstable hierarchy estimates; we down-weight those terms and flag low confidence. Finally, style choices (e.g., heavy parentheticals) can influence counts; we therefore recommend using RFI as a comparison tool across drafts, not as a single absolute bar for publication.

## 7  Broader Impact & Ethics

RFI is a pro-reader signal. We disclose features, release code, and caution against optimizing the number alone. Pair with plain-language review and user testing to avoid harmful oversimplification. Data sources in our examples (FAR) are public-domain regulatory text; no personal data are used. Optional practitioner ratings are collected with consent and without sensitive attributes; these ratings are anonymous and aggregate-only.

**Potential negative impacts.** A numeric score can incentivize "optimizing to the metric" (e.g., deleting references without adding local summaries). We mitigate by pairing RFI with edit guidance, cautioning against removing essential context, and recommending plain-language reviews and usability testing alongside RFI.

## 8  Reproducibility Statement

We release code and artifacts (JSON, briefs). Runs are deterministic; minor differences may arise from input formatting. Exact sections/paragraphs are recorded to support replication.

## 9 Conclusion

RFI offers a practical, scope-aware signal of regulatory complexity that remains interpretable for policy teams and transparent for researchers. By pairing an actionable brief with a minimal scaling check, we keep one foot in empirical rigor and the other in day-to-day usefulness.

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
