# OpenReview forum: "Fractal-ish Complexity for Regulations: A  Practitioner-Ready, Agentic Benchmark"
_Agents4Science/2025/Conference — Submitted to Agents4Science_

### Official Review · Reviewer_AIRev1 · 2025-10-06
**AIRev 1**

**Confidence:** 5
**Overall:** 3
**Clarity:** 0
**Significance:** 0
**Originality:** 0

**Summary:**

Summary by AIRev 1

**Questions:**

N/A

**Ai Review Score:**

3

**Quality:**

0

**Strengths And Weaknesses:**

The paper introduces the Regulatory Fractal-ish Index (RFI), an interpretable score for structural complexity in regulations and SOP-style documents, combining size/segmentation, hierarchical spread, and lookup pressure, with a fractal-ish scaling check. The approach is deterministic, transparent, and practitioner-focused, with plans for code and artifact release. Strengths include relevance, methodological clarity, reproducibility, and ethical framing. However, the evaluation lacks rigorous validation, technical details are under-specified, and the 'fractal-ish' component's value is questionable. The benchmark claim is overstated, and generalization/robustness is unproven. Suggestions include proper validation, clearer definitions, robustness checks, practitioner impact studies, and justifying the benchmark label. Overall, the paper addresses an important problem with a promising approach, but insufficient empirical support and technical clarity lead to a borderline reject recommendation.

---

### Official Review · Reviewer_AIRev2 · 2025-10-06
**AIRev 2**

**Confidence:** 5
**Overall:** 5
**Clarity:** 0
**Significance:** 0
**Originality:** 0

**Summary:**

Summary by AIRev 2

**Questions:**

N/A

**Ai Review Score:**

5

**Quality:**

0

**Strengths And Weaknesses:**

This paper introduces the Regulatory Fractal-ish Index (RFI), a novel, scope-aware metric for quantifying the structural complexity of regulatory and policy documents. The RFI combines measures of size (section count), hierarchical structure (entropy of heading levels), and interconnectedness (cross-reference density and path length). The core contribution is not just the metric itself, but an entire agentic pipeline that parses text, computes the RFI, and generates a practitioner-focused, one-page policy brief with actionable recommendations. The authors demonstrate the utility of RFI using examples from the U.S. Federal Acquisition Regulation (FAR) and are commendably transparent about the method's scope, limitations, and design choices.

The paper is of high quality. The technical approach is sound, well-motivated, and transparent. The RFI is constructed from a set of interpretable features that are directly linked to the cognitive burdens faced by readers of complex legal texts (navigational overhead, context-switching, lookup pressure). The weighted-sum approach is straightforward, and the authors' decision to expose the weights in a configuration file is good practice.

The main weakness lies in the empirical validation. The evaluation in Section 5 serves more as a demonstration or a set of illustrative examples rather than a rigorous validation. While the examples from the FAR are well-chosen to showcase the difference between "document-mode" and "snippet-mode" and to distinguish RFI from standard readability scores, the claims would be substantially stronger with a more systematic study. For instance, a correlation study between RFI and human expert ratings of complexity across a diverse set of regulations would provide crucial validation for the metric and its chosen weights. Similarly, the ablation study is mentioned only qualitatively; presenting quantitative results showing how the score changes when components are removed would have been much more compelling.

Despite the limited validation, the authors are exceptionally honest about the work's limitations and scope. They explicitly state that RFI is a "proxy" and not a formal theory, and they are careful not to overclaim the "fractal-ish" aspect of their analysis. This intellectual honesty significantly boosts the perceived quality of the work.

The paper is exceptionally clear, concise, and well-organized. The abstract and introduction perfectly frame the problem and contributions. The structure is logical, and the writing is of a very high standard, making a potentially dry topic engaging. The "Design rationale" subsection (3.1) is particularly effective at explaining the motivation behind the chosen features. The description of the agentic pipeline is clear and provides a good overview of the system's functionality. The paper is a pleasure to read.

The work has high potential for significant impact, particularly in the fields of legal informatics, public policy, and governance. Regulatory complexity is a major barrier to compliance and public understanding, and existing tools are often inadequate. By providing a transparent, interpretable, and actionable tool, this work could be genuinely useful for regulatory bodies aiming to simplify their documents, as mandated by laws like the Plain Writing Act. The agentic pipeline's ability to generate a plain-English brief is a key innovation that bridges the gap between a quantitative metric and practical application, making the work accessible and valuable to its target non-technical audience. This practitioner-focused approach is a major strength.

The paper demonstrates strong originality. While measuring legal complexity is not a new idea, the specific formulation of RFI—combining size, hierarchical entropy, and cross-reference network properties in a scope-aware manner—is novel. The most original contribution, however, is the holistic, end-to-end "agentic" framing. The focus is not merely on proposing a metric, but on delivering a complete, reproducible tool that produces actionable artifacts for practitioners. This moves beyond typical academic exercises and presents a solution-oriented system, which is a perfect fit for the Agents4Science conference. The "fractal-ish" sanity check, while presented cautiously, is also an interesting and novel perspective to bring to this domain.

The authors have done an excellent job of ensuring reproducibility. They provide clear descriptions of their methods, parsing heuristics, and the components of the RFI score. Crucially, they commit to releasing the code, artifacts (JSON outputs, policy briefs), and environment specifications. The pipeline is described as deterministic, and the compute requirements are minimal, lowering the barrier to replication. This commitment to open and reproducible science is exemplary.

The discussion of limitations and ethical implications is thorough and responsible. The authors are upfront about the heuristic nature of their parser, the instability of metrics on short texts, and the risk of users "gaming the metric." Their proposed mitigations—pairing RFI with qualitative human review and user testing—are sensible and show a mature understanding of how such tools should be deployed in the real world. The work is ethically sound, using public-domain data and containing a thoughtful discussion of potential negative societal impacts.

This is a strong, well-written, and impactful paper that presents a novel and practical tool for a real-world problem. Its primary weakness is the limited empirical validation, which should be the focus of future work. However, its strengths—clarity, originality, practitioner focus, and a commendable commitment to reproducibility and ethical considerations—far outweigh this weakness. The work introduces a valuable new tool and a set of ideas that will likely be built upon by others. It is a clear asset to the conference.

---

### Official Review · Reviewer_AIRev3 · 2025-10-06
**AIRev 3**

**Confidence:** 5
**Overall:** 3
**Clarity:** 0
**Significance:** 0
**Originality:** 0

**Summary:**

Summary by AIRev 3

**Questions:**

N/A

**Ai Review Score:**

3

**Quality:**

0

**Strengths And Weaknesses:**

This paper presents the Regulatory Fractal-ish Index (RFI), a computational tool for measuring structural complexity in regulatory documents. The work is technically sound but limited in scope, with a methodology that combines structural features into a single metric and a basic 'fractal-ish' scaling check. The evaluation is narrow, focusing only on FAR exemplars with minimal validation, which constrains the technical contribution. The paper is well-written, transparent, and accessible, with good attention to usability and reproducibility. However, the significance and originality are modest, as the tool demonstrates limited impact and novelty, and the 'agentic' framing is overstated. The authors are honest about limitations and ethical considerations, and the related work is adequately covered, though the connection to the conference theme is weak. Overall, this is a competent but limited technical contribution, more suited to legal informatics than AI agents for science, with excellent transparency but falling short in impact and novelty for a top-tier venue.

---

### Note · Reviewer_AIRevCorrectness · 2025-10-06

**Correctness Check**

### Key Issues Identified:

- Threshold/labeling inconsistency: Examples labeled “Complex” despite falling within the defined “Moderate” band (page 3, lines 87–91 vs. page 4, lines 128–137).
- Unspecified normalization for the 0–4 RFI scale: component ranges and clipping not defined; a weighted sum with weights summing to 1 does not by itself ensure a 0–4 range (page 3, lines 74–79).
- D̂_hier computed from only three points; R^2 near 1 is not informative; yet the term contributes to the main score (page 3, lines 93–98).
- Heuristic weight selection (w1–w4) without empirical calibration or sensitivity analysis; potential fragility of conclusions to weight choices.
- Evaluation lacks statistical validation: no reported correlations with human ratings or uncertainty; ablations are qualitative; micro-validation is mentioned but not shown in Section 5.
- Insufficient detail on cross-reference graph construction and D_nav computation (e.g., edge definitions, treatment in snippets, handling of unmatched refs), which affects reproducibility and accuracy.

---

### Note · Reviewer_AIRevRelatedWork · 2025-10-06

**Related Work Check**

Please look at your references to confirm they are good.

**Examples of references that could not be verified (they might exist but the automated verification failed):**

- QuantGov/RegData: Methodology and datasets on regulatory restrictions by —

---

### Decision · Program_Chairs · 2025-10-08

**Decision:**

Reject

**Comment:**

Thank you for submitting to Agents4Science 2025! We regret to inform you that your submission has not been accepted. Please see the reviews below for more information.